# rRNA methylation by Spb1 regulates the GTPase activity of Nog2 during 60S ribosomal subunit assembly

Kamil Sekulski [1,2], Victor Emmanuel Cruz [1,2], Christine S. Weirich [1] & Jan P. Erzberger [1] ✉

Biogenesis of the large ribosomal (60S) subunit involves the assembly of three rRNAs and 46 proteins, a process requiring approximately 70 ribosome biogenesis factors (RBFs) that bind and release the pre-60S at specific steps along the assembly pathway. The methyltransferase Spb1 and the K-loop GTPase Nog2 are essential RBFs that engage the rRNA A-loop during sequential steps in 60S maturation. Spb1 methylates the A-loop nucleotide G2922 and a catalytically deficient mutant strain ($spb1^{D52A}$) has a severe 60S biogenesis defect. However, the assembly function of this modification is currently unknown. Here, we present cryo-EM reconstructions that reveal that unmethylated G2922 leads to the premature activation of Nog2 GTPase activity and capture a Nog2-GDP-AlF$_4^-$ transition state structure that implicates the direct involvement of unmodified G2922 in Nog2 GTPase activation. Genetic suppressors and in vivo imaging indicate that premature GTP hydrolysis prevents the efficient binding of Nog2 to early nucleoplasmic 60S intermediates. We propose that G2922 methylation levels regulate Nog2 recruitment to the pre-60S near the nucleolar/nucleoplasmic phase boundary, forming a kinetic checkpoint to regulate 60S production. Our approach and findings provide a template to study the GTPase cycles and regulatory factor interactions of the other K-loop GTPases involved in ribosome assembly.

Ribosomal RNA maturation includes the extensive post-transcriptional modification of nucleotides that cluster in functionally important regions of the ribosome[1,2]. In *S.cerevisiae*, snoRNPs methylate 54 nucleotides at the 2´-OH position to create 2´-O-methyl-riboses and isomerize 47 uridine bases to form pseudouridines, while dedicated enzymes catalyze the remaining 15 alterations[3]. Although their individual roles are not well defined, these nucleotide modifications collectively expand the rRNA chemical repertoire to modulate the assembly, stability and function of ribosomes[1,4–6]. Among these rRNA-directed catalytic activities, a subset are carried out during late steps of ribosome maturation rather than co-transcriptionally when rRNA access is most unencumbered. These modifications in particular have

been proposed to affect assembly events rather than the stability or function of mature ribosomes[7–10].

One such late modification is the 2´-OH methylation of G2922 (mG2922) within the A-site loop of rRNA helix 92 (H92) by the S-adenosyl-methionine (SAM)-dependent methyltransferase Spb1 (Fig. 1a, b)[7,8,11]. Notably, mG2922 is the only eukaryotic ribose methylation carried out in a snRNP-independent manner by a dedicated enzyme. Spb1 is an essential structural component of nucleolar pre-60S intermediates[11,12] and strains carrying a mutation ($spb1^{D52A}$) that abolishes G2922 methylation activity display a severe growth defect and impaired 60S biogenesis[7,8]. After Spb1 release and L1 stalk rearrangement[13], the K⁺-dependent GTPase Nog2 engages the A-loop

[1]Department of Biophysics, UT Southwestern Medical Center, 5323 Harry Hines Blvd., ND10.104B, Dallas, TX 75390-8816, USA. [2]These authors contributed equally: Kamil Sekulski, Victor Emmanuel Cruz. ✉e-mail: jan.erzberger@utsouthwestern.edu

during nucleoplasmic 60S maturation steps (Fig. 1c, d)[14–17]. Nog2 is an essential hub enzyme that acts as a key quality control element during the nucleoplasmic maturation of 60S, anchoring both the pre- and post-rotation states of the 5S rRNP (Nog2[pre] and Nog2[post]) via extensive rRNA and RBF interactions[13,14]. GTP hydrolysis by Nog2 is required for its dissociation and for the subsequent recruitment of the export factor Nmd3, which enables the nuclear export of pre-60S particles[18–21].

Here, we present cryo-EM reconstructions showing that failure to methylate G2922 causes the premature activation of Nog2 GTPase activity. A structure of the Nog2-GDP-AlF$_4^-$ transition state is consistent with a direct role for G2922 in the activation of Nog2. Genetic suppressors and in vivo imaging of the ribosomal protein uL23-GFP indicate that premature GTP hydrolysis prevents the efficient binding of Nog2 to early nucleoplasmic 60S intermediates, resulting in a 60S

**Fig. 1 | Disruption of Spb1 methylation of G2922 results in premature Nog2 GTP hydrolysis. a** Position of the methyltransferase Spb1 in the State E2 (pdb-7NAC) cryo-EM reconstruction[11]. **b** detail of the interaction between the methyltransferase domain of Spb1 (teal cartoon) and helix 92 (orange). mG2922 is located within the A-loop at the apex of H92 (boxed inset). **c** Location of the Nog2 GTPase in the cryo-EM reconstruction of the Nog2[pre] state. **d** Detail of the interaction between the GTPase domain of Nog2 and H92. mG2922 is intercalated near the active site of Nog2, protruding away from the A-loop (boxed inset). **e** Top – Surface rendition of the Nog2 GTPase active site from the *SPB1* strain (K-loop residues are omitted for clarity), showing stick models of GTP and the A-loop with intercalated nucleotide mG2922 as well as the position of catalytic ions. Bottom – Experimental maps from Nog2[pre] (*SPB1*) carved around GTP (left) and mG2922 (right), showing clear density for the γ-phosphate and the 2′-O-methyl group. **f** Top – Surface rendition of the Nog2 GTPase active site derived from the *spb1[DS2A]* strain (K-loop residues are omitted for clarity), showing stick models of GDP and catalytic ions. Bottom – Map density carved around GDP (left) and the area normally occupied by the A-loop (right), showing that GTP hydrolysis has taken place and that the A-loop (reference position shown as a transparent outline) is no longer intercalated into the active site. **g** Mg[2+] and K[+] coordination network in the GTP-bound bound active site (*SPB1* strain). Nog2 residues involved in metal coordination are shown as sticks and labeled. **h** Mg[2+] and K[+] coordination network in the GDP-bound bound active site (*spb1[DS2A]* strain). Nog2 residues involved in metal coordination are shown as sticks and labeled.

biogenesis defect. Based on these results, we propose a model in which G2922 methylation levels regulate Nog2 recruitment to the pre-60S near the nucleolar/nucleoplasmic phase boundary, forming a kinetic checkpoint to regulate 60S production.

## Results

### G2922 methylation modulates Nog2 GTP hydrolysis

Because of the proximity of mG2922 to the active site of Nog2 (Fig. 1d), we investigated the effect of G2922 methylation status on Nog2 engagement of pre-60S intermediates by obtaining high-resolution cryo-EM reconstructions of Nog2[pre] intermediates from *SPB1* (2.3 Å) and *spb1[DS2A]* (2.4 Å) strains using dual affinity tags on Nog2 and the biogenesis factor Tif6 (Supplementary Figs. 1–4). While there are no major differences in RBF composition compared to earlier structures, the ~0.7 Å improvement in the overall resolution of our maps compared to previous reconstructions of Nog2[pre] enabled us to build an improved atomic model that includes all observable rRNA nucleotide modifications, ordered Mg[2+] ions and their coordinated waters as well as three ordered Bis-Tris-Propane molecules (Supplementary Table 1 and Supplementary Fig. 5). Our model corrects a number of register shifts and mis-paired rRNA bases as well as improving the completeness of individual RBF and RP models. In the active site of Nog2 reconstructed from the *SPB1* strain, we were able to precisely model GTP, Mg[2+] and K[+] ions, correcting the GTP orientation from earlier Nog2[pre] models (Fig. 1e and Supplementary Fig. 6a–d)[14]. There is clear density for the methyl group of mG2922 intercalated in near proximity (~7.2 Å) to the γ-phosphate of the GTP (Fig. 1e). In Nog2[pre] intermediates purified from the *spb1[DS2A]* strain, Nog2 occupies an identical position and maintains all of its RBF, RP and rRNA interactions, but the Nog2 active site is strikingly different (Fig. 1f): There is no density corresponding to the GTP γ-phosphate, indicating that GTP hydrolysis has taken place and that GDP occupies the Nog2 active site. Additionally, unmethylated G2922 is no longer intercalated near the Nog2 active site and the A-loop is only weakly ordered (Fig. 1f). The high resolution of our maps allowed us to infer the coordination network in the active sites of the GDP and GTP-bound states (Fig. 1g, h) and to define the ~1 Å shift in the orientation of the K-loop that occurs after GTP hydrolysis (Supplementary Fig. 6e). These structures indicate that the methyl group of mG2922 acts as an inhibitor of Nog2 GTP hydrolysis in the *SPB1* wild-type strain. This inhibition could be either direct, if the absence of the methyl group allows G2922 to directly trigger hydrolysis, or indirect, if the intercalated mG2922 nucleotide blocks active site access by a different *cis* or *trans*-activating element.

### G2922 is positioned to directly activate Nog2

To address this question, we set out to capture the transition state of Nog2 GTPase activation by adding AlF$_4^-$ to Nog2[pre] intermediates purified from the *spb1[DS2A]* strain. Because the disordered A-loop remains in the vicinity of the active site after premature GTP hydrolysis, we hypothesized that addition of AlF$_4^-$ could recapitulate a GTP hydrolysis transition state in these particles (Fig. 2a). A 2.4 Å cryo-EM reconstruction of this sample (Supplementary Figs. 3, 7) shows

additional planar density consistent with AlF$_4^-$ in the active site of Nog2 as well as the reappearance of density for G2922 and the rest of the A-loop near the Nog2 active site (Fig. 2b). As expected, the 2′-OH of G2922 is now unmethylated and helps co-ordinates a triad of water molecules linking G2922 and AlF$_4^-$ (Fig. 2b, c) The water closest to the AlF$_4^-$ moiety is at a distance of ~2.0 Å, consistent with the predicted position of the attacking nucleophilic water. This catalytic water is coordinated by the backbone carboxyl group of residue T350 as well as a bridging water that forms a hydrogen bond with the C2 amino group of the G2922 base and the backbone carbonyl of G369 within sensor II. The third water is H-bonded to the 2′OH of the G2922 ribose, assuming a position similar to that occupied by the methyl group of mG2922 in the WT *SPB1* structure (Figs. 1e, 2c, Supplementary Movie 1). The Nog2 transition state shows a remarkable structural similarity to the high-resolution crystal structure of the bacterial K-loop GTPase MnmE[22], with near identical positioning of GDP, AlF$_4^-$ and the catalytic water network (Fig. 2c, d). We therefore propose that unmethylated G2922 acts as a *trans*-activating factor, organizing and activating a water network to trigger the nucleophilic attack on the GTP γ-phosphate. We believe this represents the first evidence of direct activation of a GTPase by an RNA nucleotide.

### Effect of early GTP hydrolysis on Nog2 pre-60S intermediates

The overall structure of the Nog2[pre] intermediate is not affected by early Nog2 GTP hydrolysis, indicating the γ-phosphate is not required to maintain Nog2 binding. Further, we were also able to reconstruct the Nog2[post] intermediate state, both with and without the Rix1 complex[13], from a subclass of particles purified from the *spb1[DS2A]* strain (Fig. 3a and Supplementary Figs. 3, 4). Refinement of the Rix1-containing subset to 2.9 Å resolution shows that GDP-bound Nog2 remains associated with the 60S during 5S rotation and Rix1 complex binding, as the Nog2 active site in this reconstruction also contains GDP (Fig. 3b) and lacks clear density for the A-loop. Just like Nog2[pre], Nog2[post] is not destabilized by the presence of Nog2-GDP, indicating that 5S rotation, Rix1-complex binding and Rea1 recruitment can proceed normally in the context of Nog2-GDP. Because the presence of Nog2-GDP did not cause any structural defects or instabilities in late nucleoplasmic intermediates, we instead hypothesized that the 60S biogenesis defect of the *spb1[DS2A]* strain arises during the nucleolar/nucleoplasmic transition of pre-60S particles. This transition is initiated by the removal of a set of RBFs by the AAA+ATPase Rea1, triggering a rearrangement of the L1 stalk that allows it to dock onto the pre-60S core[11,13] (Fig. 3c). These conformational changes expose the Nog2 docking site and permit its peripheral tail interactions. The presence of weak density for the Nog1-interaction loop of Nog2 in the NE1 structure[11] suggests that Nog2 is initially loosely tethered to the pre-60S until the GTPase domain can dock. X-ray structures of isolated RbgA[23,24], the bacterial homolog of Nog2 that engages an analogous position in pre-50S bacterial ribosomes, reveal that the K-loop of these enzymes does not adopt an ion-coordinating, mature conformation before pre-ribosome binding (Supplementary Fig. 8a). In the Nog2[pre] structure, the K-loop is stabilized by a conserved

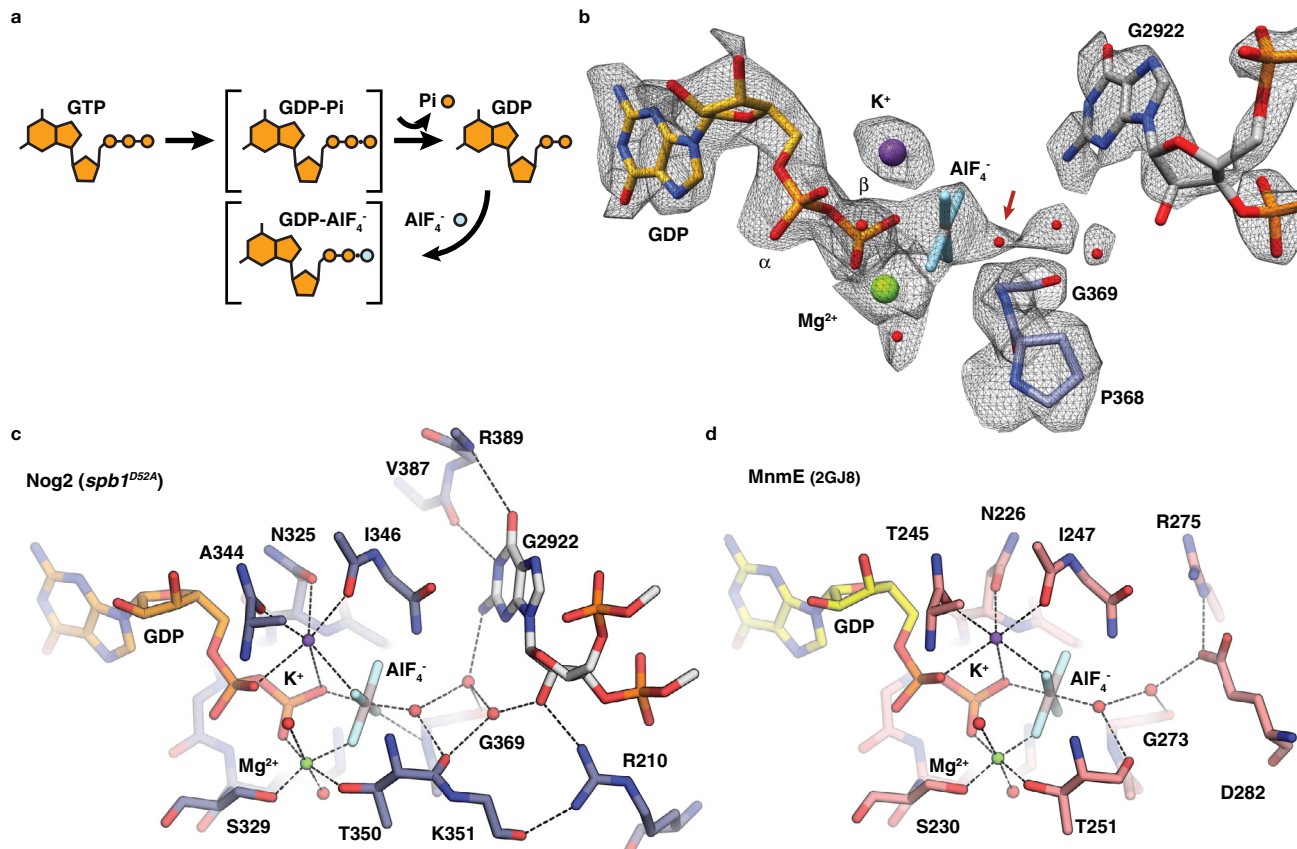

**Fig. 2 | G2922 is directly involved in Nog2 GTPase activation. a** Schematic of the transition state trapping strategy. $AlF_4^-$ was added back to Nog2[pre] intermediates purified from the *spb1DS2A* strain to mimic the transition state of GTP hydrolysis. **b** Map density in the active site of Nog2 after $AlF_4^-$ addition, showing extra density consistent with the presence of tetragonal $AlF_4^-$ and unmethylated G2922. **c** Active site coordination in the reconstituted transition state structure showing the organized water triad. Nog2 residues participating in the coordination network are

shown as sticks. **d** The active site of the archetypal K-loop GTPase MnmE[22], solved at 1.7 Å resolution in the presence of GDP·$AlF_4^-$, showing a near-identical coordination structure to that observed in the Nog2 transition state. The coordination of the attacking water, which is carried out by protein side chains in MnmE, is also highly similar. MnmE residues participating in the coordination network are shown as sticks.

interaction with H64 (Supplementary Fig. 8a). H64 and H92 therefore promote the initial binding of Nog2 and combine to organize its active site. We hypothesize that premature activation of Nog2 disrupts this process. Stable Nog2[pre] intermediates can nevertheless still form if secondary binding interactions involving the Nog2 N-terminal tail can form to compensate for the loss of the A-loop interaction by Nog2-GDP. Reasoning that the reduced formation of Nog2[pre] particles during that nucleolar/nucleoplasmic transition could be overcome by increasing the concentration of Nog2 in vivo, we overexpressed *NOG2* in the *spb1DS2A* strain background, partially rescuing the growth defect (Fig. 3d). GTP hydrolysis is still required for growth rescue, since overexpression of *nog2[G369A]*, which binds but cannot hydrolyze GTP[19], fails to rescue *spb1DS2A* growth (Fig. 3d). This rescue was also specific to *NOG2*, since overexpression of *SPB1-MTD[NOLS]* or *NMD3* failed to rescue growth (Supplementary Fig. 8b, c). Unlike previous studies[19], overexpression of *nog2[G369A]* in the *SPB1* strain is not dominant negative (Fig. 3d), likely due to differences in strain backgrounds and overexpression systems.

**Intragenic suppressors bypass the growth defect of *spb1[DS2A]***
Additional evidence for this model comes from two spontaneous intragenic suppressors of *spb1[DS2A]* (*spb1[K767N]* and *spb1[E769K]*), isolated during our genetic studies. Both mutations lead to a near-complete rescue of the growth defect of *spb1[DS2A]* cells (Fig. 4a). To ensure that the mutations do not simply rescue the catalytic activity of Spb1, we

obtained a cryo-EM reconstruction of Nog2[pre] purified from the *spb1[DS2A/E769K]* double-mutant strain to 2.4 Å resolution (Supplementary Figs. 2, 7). Although these cells grow nearly as well as *SPB1* wild-type cells, Nog2 is GDP-bound (Fig. 4b), showing that the suppressor mutations rescue the growth defect of *spb1[DS2A]* despite the premature activation of Nog2 GTP hydrolysis. The suppressor mutations map to the structural, C-terminal portion of Spb1 (Fig. 4c) and insights into how these mutations affect the binding kinetics of Nog2 can be obtained from our recent structure of the NE1 pre-60S intermediate that captures the Spb1-Nog2 transition intermediate state (Fig. 3c)[11]. We were able to dock a structural model for the helical structure containing the suppressors using low-pass filtered maps and information from Alphafold[25] models for this segment of Spb1 (Fig. 4d, e and Supplementary Fig. 8d, e). Both suppressors map to the turn at the apex of two V-shaped helices that extend into the region surrounding the disengaged H92 helix of state NE1)[11]. Modeling of the Nog2 GTPase domain onto this structure shows that it sterically clashes with the helical structure containing the suppressors (Fig. 4f). Thus, binding of the Nog2 GTPase domain cannot occur until the Spb1 helical element vacates the region around H92. We hypothesize that the suppressor mutants destabilize the helical structure in the C-terminal portion of Spb1, vacating the region around h92 to allow early, unfettered A-loop access, improving the overall binding kinetics of the Nog2 GTPase domain. The suppressors may facilitate the formation of secondary Nog2 binding interactions by modulating the kinetics of the L1 stalk

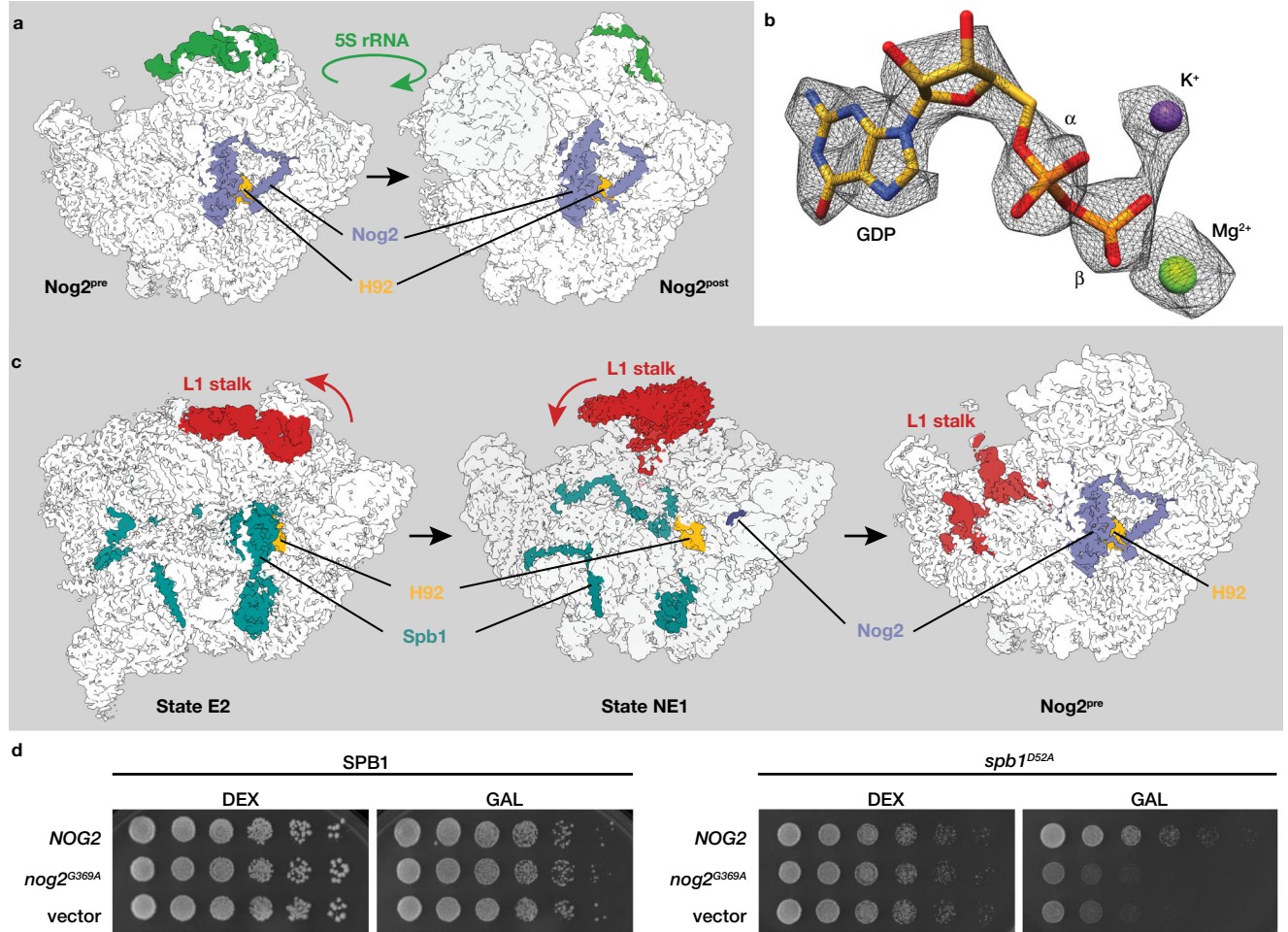

**Fig. 3 | Premature GTP hydrolysis does not affect late nucleoplasmic 60S maturation but prevents the efficient formation of early nucleoplasmic intermediates. a** Low-pass-filtered maps (4.5 Å) of Nog2$^{pre}$ and Nog2$^{post}$, purified from the *spb1$^{D52A}$* strain highlighting the position of Nog2 (blue) and 5S rRNA (green) in its pre- and post-rotation states. **b** Map density in the Nog2 active site from Nog2$^{post}$ particles purified from the *spb1$^{D52A}$* strain, showing clear occupancy by GDP. **c** Low pass filtered maps (4.5 Å) of pre-60S structures showing the transition between Spb1 and Nog2-containing intermediates near the nucleolar/nucleoplasmic boundary. The transition is precipitated by the catalytic activity of the ATPase Rea1 and is marked by the movement and docking of the L1 stalk (shown in red) and the formation of an organized 5S rRNP. In the NE1 state (pdb-7U0H), the C-terminal portion of Spb1 (teal) is still bound, blocking L1 rearrangement. Initial engagement of Nog2 also occurs at this juncture, as density for the Nog2 helix (blue) near the RBF Nog1 is also observed. **d** Overexpression of *NOG2* partially rescues the slow growth phenotype of the *spb1$^{D52A}$* strain. Plasmids containing Gal-inducible *NOG2*, *nog2$^{G369A}$* or empty vector were transformed into *SPB1* or *spb1$^{D52A}$* strains and plated on selective synthetic media containing glucose (DEX) or galactose (GAL). Overexpression of *NOG2*, but not the catalytic mutant *nog2$^{G369A}$* partially rescued the growth defect of *spb1$^{D52A}$* cells.

docking. A kinetic basis for the defect observed in the *spb1$^{D52A}$* strain is also suggested by the fact that the growth defect is partially rescued at 37 °C (Fig. 4a)[7].

### Nucleolar/nuclear accumulation of pre-60S in *spb1$^{D52A}$* cells

Both the suppressor mutants and the temperature-dependence of the *spb1$^{D52A}$* phenotype suggest that modulation of Nog2 binding affects pre-60S assembly kinetics. Therefore, we visualized ribosome biogenesis in vivo by expressing GFP-tagged ribosomal proteins to monitor the assembly of large (uL23-GFP) or small (uS7-GFP) ribosomal subunits[26–28]. Saturated overnight cultures of *SPB1*, *spb1$^{D52A}$* or *spb1$^{D52A/E769K}$* all show an even cytoplasmic distribution of ribosomal subunits (Fig. 5a, b), but upon resumption of ribosome biogenesis after dilution into fresh media, nuclear accumulation of pre-60S intermediates, but not pre-40S, is observed in ~2/3 of cells of the *spb1$^{D52A}$* strain (Fig. 5a, b). Some accumulation (~15% of cells) is also observed in the *spb1$^{D52A/E769K}$* strain, especially at the 1 and 3 hour post-dilution timepoints, consistent with our hypothesis that the reduced binding of prematurely activated Nog2 represents a kinetic defect in the *spb1$^{D52A}$*

strain that can be partially overcome in the suppressor background. To confirm that the GFP signal reflects the accumulation of pre-60S particles and not free uL23-GFP, we carried out semi-quantitative mass spectrometry analysis of Tif6-containing pre-60S intermediates purified from either *SPB1* and *spb1$^{D52A}$* strains (Fig. 5c, Source data). Tif6 binds the pre-60S in the nucleoplasm and remains bound until final maturation in the cytoplasm, allowing for global changes in pre-60S distribution to be monitored[10,20]. A comparison of normalized abundances of RBFs and RPs shows an accumulation of RBFs that first engage the pre-60S in the nucleolus and a reduction of RBFs that engage the pre-60S during late nucleoplasmic and cytoplasmic maturation (Fig. 5d). Our structural, genetic, and localization data are in agreement with a recently published study on the function of G2922 methylation during 60S assembly[29].

### Discussion

Activation of Nog2 GTPase activity promotes its release from the pre-60S and allows the efficient recruitment of the export factor Nmd3. Our finding that the *spb1$^{D52A}$* suppressor mutations sustain near wild-type

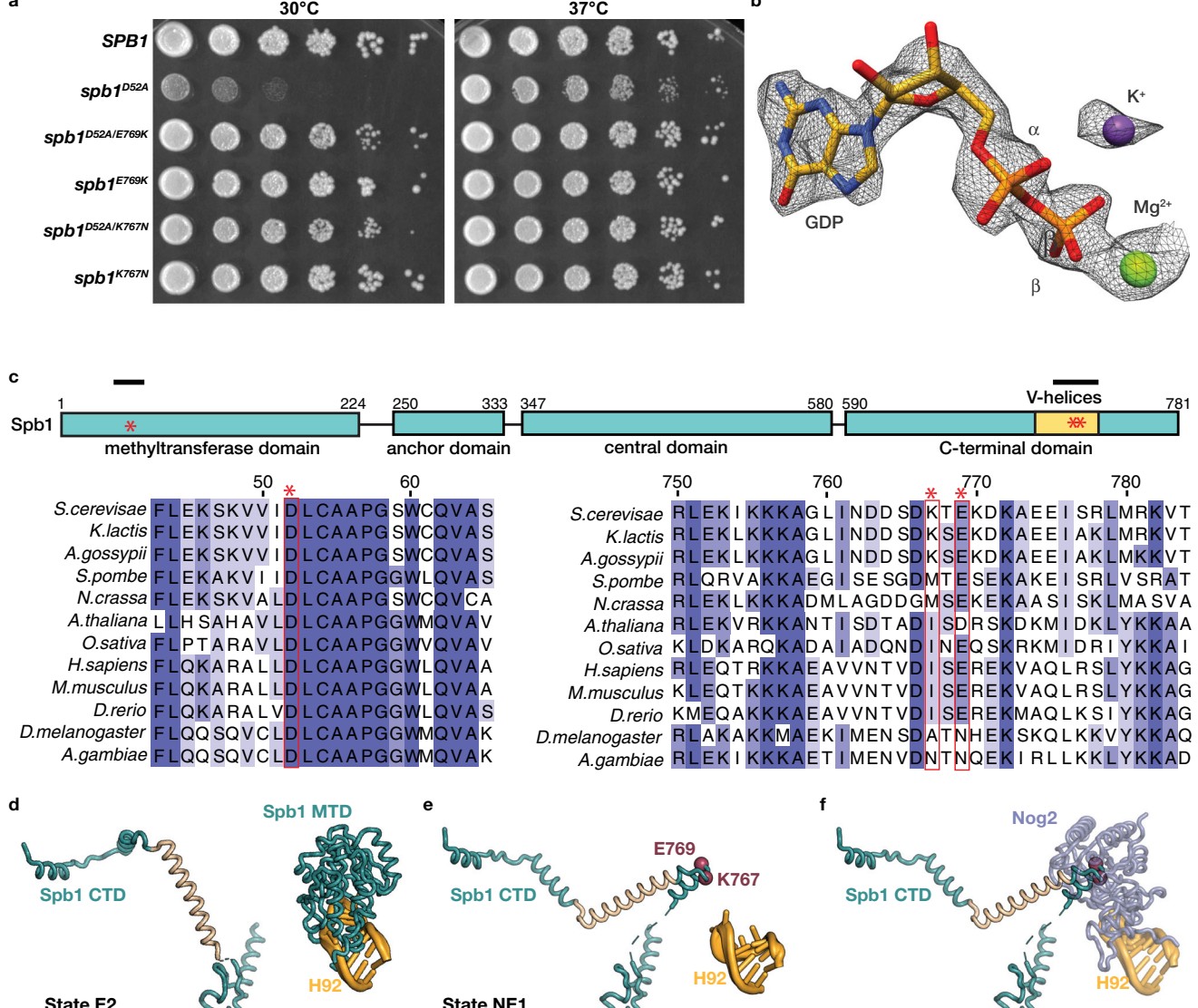

**Fig. 4 | Suppressor mutations in the C-terminal domain of Spb1 restore growth of *spb1*^DS2A in the absence of G2922 methylation. a** Spontaneous intragenic suppressors of *spb1*^DS2A rescue the slow growth phenotype of *spb1*^DS2A cells at 30 °C but do not influence cell growth on their own. As previously reported, growth of *spb1*^DS2A is restored at 37 °C. **b** Map density in the Nog2 active site from Nog2^pre particles purified from the *spb1*^DS2A/E769K strain, showing clear occupancy by GDP and indicating that normal 60S nucleoplasmic maturation occurs in the presence of Nog2-GDP. **c** Domain schematic (top) of Spb1 showing the major segments of the protein as well as the position of D52 and of the suppressor mutations (red stars).

The position of the V-helices is colored in yellow and the segments shown as sequence alignments are highlighted by black bars. (Bottom) Sequence alignments of the regions flanking the suppressor mutations are colored by the level of conservation per residue. **d** Cartoon depiction of the Spb1-CTD in the E2 and **e** NE1 states, showing the structural rearrangement of the central helical element (cream) and the release of the Spb1-MTD from H92. The position of the suppressor mutants at the apex of the v-shaped helices is shown as red spheres. **f** The helical element extends into the vicinity of H92, sterically blocking the Nog2 GTPase binding site.

growth even though Nog2 is GDP-bound throughout the nucleoplasmic maturation of the large subunit shows that although hydrolysis is necessary for Nog2 release, these events can be uncoupled without affecting the timing of Nog2 release and Nmd3 binding. This suggests that at a specific point in late nucleoplasmic maturation, after Rix1-Rea1 mediated removal of Rsa4, Nog2 binding to the pre-60S depends on the presence of the GTP γ-phosphate. How the GTPase activity of Nog2 is activated during normal maturation remains undefined, but potential mechanisms include a conformational change at the Nog2/H92 interface that pivots mG2922 in a way that cancels the inhibitory effect of the methyl group without affecting the ability of the C2-amine of mG2922 to coordinate the catalytic water network. Alternatively, Rsa4 removal could trigger the release of H92, allowing catalytic pocket access to a different Nog2 trans-activating element.

The direct link between Spb1 methylation and Nog2 GTPase activation raises the question about possible biological functions of premature Nog2 activation and the circumstances under which it is triggered in cells. One intriguing possibility is that Spb1 acts as a sensor to reduce the rate of 60S biogenesis in response to limiting intracellular concentrations of SAM. The concept of methyltransferases acting as SAM-sensors is well established in numerous regulatory pathways[30,31]. This model explains why G2922 methylation is carried out by a specialized enzyme rather than a snoRNP. Intriguingly, the *E.coli* homolog of Spb1, RlmE, is the only bacterial rRNA-directed methyltransferase that displays reduced catalytic activity in vivo when intracellular SAM levels are reduced[32]. We propose that under conditions of limiting SAM, Spb1 is outcompeted for the reduced intracellular SAM pool by other cellular methyltransferases, notably Nop1, the

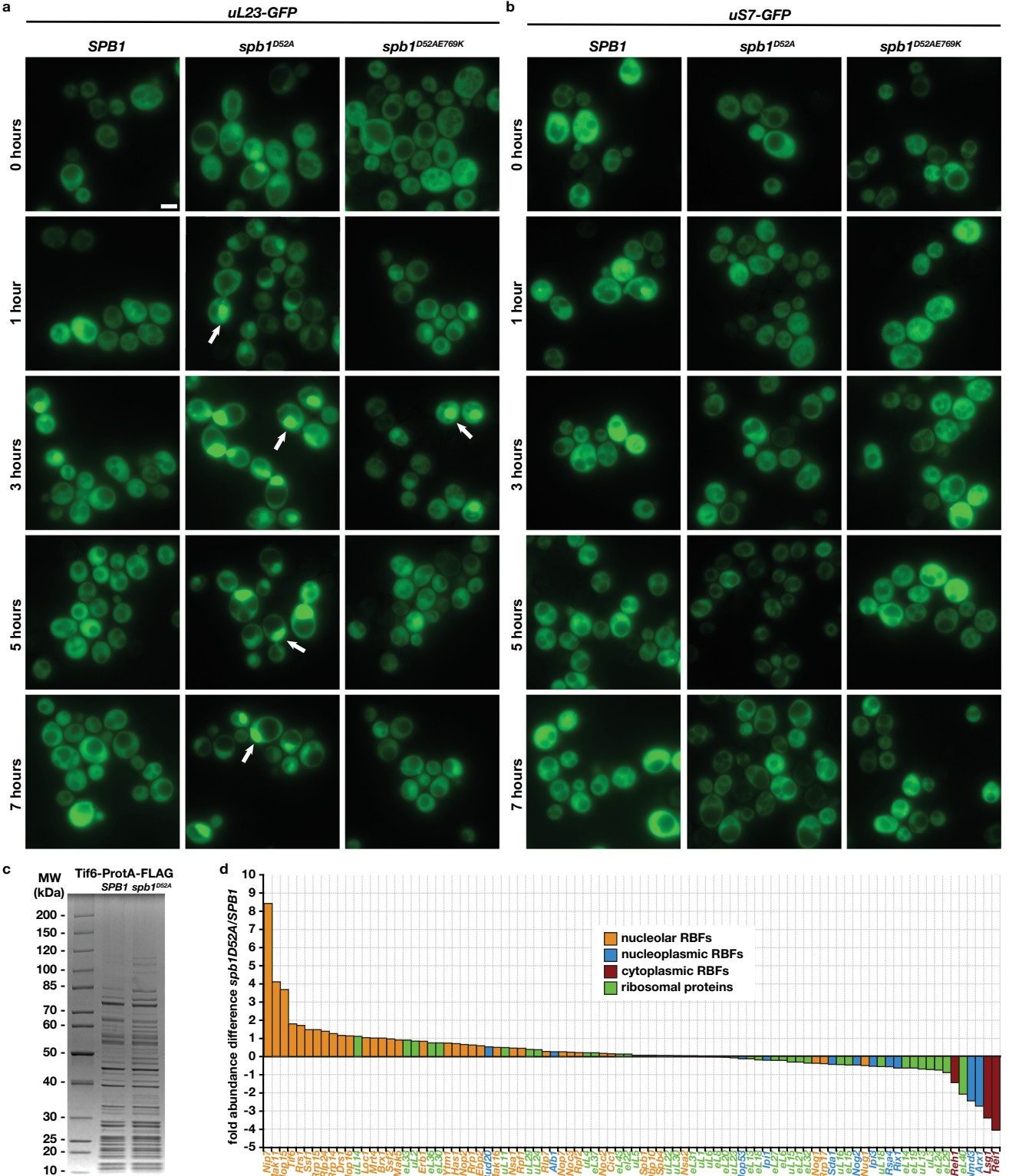

**Fig. 5 | Nuclear accumulation of pre-60S intermediates.** Time course of the in vivo distribution of GFP-tagged large (**a**) and small (**b**) ribosomal proteins in *SPB1*, *spb1^D52A^*, and *spb1^D52A/E769K^* cells after dilution of a saturated culture into fresh media (at t = 0 hours). White arrows indicate representative cells with nuclear accumulation of GFP signal. Images are pseudo-colored and scale bars represent 5 μm. **c** Coomassie-stained SDS-PAGE gel of tandem-purified Tif6-containing pre-60S intermediates from *SPB1* and *spb1^D52A^* cells. Identical samples were analyzed by semi-quantitative mass spectrometry. **d** Visualization of semi-quantitative mass-spectrometry data sorted by differences in abundance between *spb1^D52A^* and *SPB1* strains using representative RBFs and RPs between the two samples. RBFs are colored according to the subcellular compartment where they first bind pre-60S intermediates, showing a clear trend of nucleolar RBFs enrichment and late nucleoplasmic and cytoplasmic RBFs reduction. RP abundances are shown in green.

highly abundant catalytic component of the box C/D snoRNP complex. Failure to bind SAM does not affect the essential structural functions of Spb1 in nucleolar 60S maturation, but the subsequent intercalation of unmodified G2922 in the active site of Nog2 triggers premature GTP hydrolysis, resulting in the kinetic trapping of pre-60S intermediates at the nucleolar/nucleoplasmic boundary. Such a mechanism would allow an immediate and proportional reduction of 60S biogenesis in response to reduced cellular levels of SAM.

## Methods

### Yeast strains
Yeast strains used in this study are listed in Supplementary Table 2, plasmids used in this study are listed in Supplementary Table 3. Genomic knockout strains in yeast were performed using standard PCR-mediated gene disruptions[33].

### Microscopy
For localization of uL18-GFP and uS5-GFP, strains were grown overnight in synthetic complete -uracil (SC-URA) medium with 2% Raffinose, and diluted 10x in SC-URA + 2% Raffinose and 1% Galactose. At each time point, cells were mounted in 1% low melt agarose and imaged immediately. Fluorescence micrographs were recorded on an inverted Nikon EclipseTi microscope fitted with a 100X Plan Apo objective and a Andor Zyla 5.5 sCMOS camera. Images were processed after acquisition by contracting signal ranges and pseudo-coloring. For localization of Spb1-MTD variants, grown overnight in synthetic complete -uracil (SC-URA) medium with 2% Raffinose, and diluted to $OD_{600}$ of 0.1 in SC-URA + 2% Raffinose and 1% Galactose to an OD of 0.5, mounted in 1% low melt agarose and imaged immediately. Images were taken on a Nikon Ti2-E inverted microscope equipped with a Yokogawa CSU-X1 spinning disk and a 100x NA = 1.49 oil objective. Samples were illuminated with a 488 nm solid state laser light source and images collected on an ORCA-FLASH 4.0 sCMOS camera. A z-series of confocal images was acquired for two fields of view in a single biological replicate. Image processing was performed using ImageJ.

### Isolation and identification of a spb1^DS2A suppressors
Fast growing colonies that spontaneously arose from the $spb1^{DS2A}$ strain were isolated and back-crossed to $spb1^{DS2A}$ to confirm the presence of a single mutation. Sequencing of candidate genes revealed the single point mutations ($spb1^{K767N}$ and $spb1^{E769K}$).

### Overexpression and purification of Nog2 containing pre-ribosomes
For yeast overexpression, full length wild-type *NOG2* was cloned into a modified pRS406 plasmid that introduced a C-terminal MYC–TEV–2xStrep tag under the control of a β-estradiol inducible system[34]. Tif6 was tagged at the genomic locus with a C-terminal 3xFLAG-3C–2xProteinA. Starter cultures for each strain were grown overnight in YPD to saturation. The following morning 12 L of YPD were inoculated from the starters to an $OD_{600}$ of 0.1 and grown at 30 °C to an $OD_{600}$ of 0.8. Overexpression was induced for 1 hour by the addition of β -estradiol to a final concentration of 2 µm. Cells were harvested by centrifugation for 20 min at 4000 g, the pellets were washed in Ribo-buffer A (50 mM Bis-Tris-Propane HCl pH 8.0, 125 mM NaCl, 25 mM KCl, 10 mM $MgCl_2$, 1 mM TCEP and 0.1% (w/v) NP-40) and centrifuged again for 10 min at 4000 g. Cell pellets were harvested and flash frozen in liquid nitrogen. Cryogenic lysis was done using a grinding ball mill (Fritsch Pulverisette 6). For purification of pre-ribosomes, 30 g of lysate are warmed to 4 °C before addition of Ribo-buffer A supplemented with E64, pepstatin, PMSF and RNAse-free DNase I. Lysate was cleared by centrifuging at 100,000 g for 30 min and loaded onto IgG Sepharose resin (Cytiva). Samples were washed with 10 column volumes of Ribo-buffer A followed by 10 column volumes of Ribo-buffer B (50 mM Bis-Tris-Propane HCl pH 8.0, 125 mM NaCl, 25 mM

KCl, 10 mM $MgCl_2$, 1 mM TCEP and 0.01% (w/v) NP-40). Protein-A tags were cleaved on-column using 3 C protease and eluates were applied to a Strep-Tactin column (Cytiva), washed with 5 column volumes of Ribo-buffer B and eluted using Ribo-buffer B supplemented with 10 mM desthiobiotin. Eluted pre-ribosomes were concentrated on Amicon Ultra 0.5 ml spin columns with a 100 kDa cutoff (Merck Millipore). Starter strains carrying spb1^DS2A were grown for 36 h to reach saturation and 12 L cultures were grown to 12 – 16 hours to reach an $OD_{600}$ of 0.8. A fraction of the Nog2 • Tif6 intermediates obtained from the $spb1^{DS2A}$ strain were incubated with 50 µM $AlF_4^-$ for 30 minutes before preparation of cryo-EM grids.

### Cryo-EM grid preparation and Data Collection
Cryo-EM grids were prepared using a Mark IV Vitrobot (FEI) set at 4 °C and 100% humidity by applying 3 µL of Nog2 • Tif6 samples at a concentration of 8 $A_{260}ml^{-1}$ (260 nM) to the carbon side of glow-discharged continuous carbon coated Quantifoil R2/1 300-mesh grids. After sample application a wait time of 10 s was used, followed by grid blotting with a blot force of 13 and blot time of 4.5 s. After blotting, grids were quickly plunged and frozen in liquid ethane. Nog2 • Tif6 from the $spb1^{DS2A}$ strain with and without $AlF_4^-$ were prepared at a concentration of 3 $A_{260}ml^{-1}$ (100 nM). The Nog2 • Tif6 sample from the suppressor strain $spb1^{DS2A/E769K}$ was prepared at a concentration of 8 $A_{260}ml^{-1}$ (260 nM). Micrographs were acquired on a Titan Krios (FEI) operated at 300 kV equipped with a Gatan K3 direct electron detector using a slit width of 30 eV on a GIF-Quantum energy filter. Automated data collection was performed using SerialEM[35] with a defocus range of −0.9 to −2.2 µm and a pixel size of 1.08 Å. Each micrograph was dose fractionated into 40 frames of 0.05 s each under a dose rate of 26.2 e⁻/pixel/s with a total exposure time of 2 seconds and a dosage of approximately 56.4 e⁻/pixel. 5580 movies were collected for the Nog2 • Tif6 sample from the *SPB1* strain, 5434 movies for the Nog2 • Tif6 sample from the $spb1^{DS2A}$ strain and 4349 movies were collected for Nog2 • Tif6 sample from the $spb1^{DS2A,E769K}$ strain. The Nog2 • Tif6 • $AlF_4^-$ sample from the $spb1^{DS2A}$ strain was collected at the PNCC using a Titan Krios operated at 300 kV with a pixel size of 1.0688 Å. 7614 movies were collected for this dataset.

### Cryo-EM image processing
Motion correction was performed using MotionCorr2[36], and CTF parameters were estimated using GCTF[37]. Movie selection was based on the quality of Thon rings and on the absence of artifacts that arise from poor ice, for the Nog2•Tif6 (*SPB1*) dataset 4910 movies were retained out of 5580 that were collected, for Nog2•Tif6 ($spb1^{DS2A}$) dataset 5327 movies were retained from an original 5434, for the Nog2•Tif6•AlF4- ($spb1^{DS2A}$) dataset 7590 movies were selected from 7614, and for the Nog2•Tif6 ($spb1^{DS2A, E769K}$) dataset 3774 movies were used from the original 4349. All subsequent image processing was performed using RELION[38].

### Nog2•Tif6 (*SPB1*) dataset
About 300 particles were picked from 3 micrographs to generate initial 2D classes to use as templates for automated particle picking from the 4910 micrographs. 1,680,783 particles were extracted, binned 4 times and used for 2D classification, from these 1,639,317 particles were selected for 3D classification. Four identical Nog2^pre classes were combined and re-extracted at the original pixel size of 1.08 Å, resulting in the total of 1,358,921 particles. 3D refinement was performed with an imposed symmetry of C1 and resulted in a reconstruction with a resolution of 2.84 Å. A mask surrounding H68 was used to subtract core density followed by a local 3D classification without re-alignment in order to select particles with strong density for H68. From this, 1,159,206 particles were selected for the final reconstruction. After another cycle of 3D refinement and postprocessing the overall resolution improved to 2.34 Å. Using a similar procedure,

another local classification was performed from the initially picked 3D classes that focused on the 5S rRNP and resulting in a 2.63 Å resolution map, notably improving the map quality for that region. For model building, the local 5S rRNP map was aligned and resampled onto the overall map grid using the VOP command in UCSF Chimera, the 5S was built into this aligned local 5S rRNP map. The final resolution in each case was estimated by applying a soft mask over the maps and was calculated using the gold-standard Fourier shell correlation (FSC) = 0.143[39]. Local resolution maps were generated using the ResMap[40] wrapper within Relion 3.1.

### Nog2•Tif6 (*spb1$^{DS2A}$*) dataset

Initial 2D classes were obtained the same way as for Nog2•Tif6 dataset. 925,065 particles were extracted after automated particle picking, binned 4 times and used for 2D classification. 905,023 particles were selected for subsequent 3D classification. Two Nog2$^{pre}$ classes were combined and re-extracted at the original pixel size of 1.08 Å, resulting in the total of 328,470 particles, the excluded pre-rotation class had much weaker density for various regions within the core and foot region. 3D refinement was performed with an imposed symmetry of C1 and resulted in an initial reconstruction with a resolution of 2.95 Å. After CTF refinement, another cycle of 3D refinement the overall resolution improved to 2.44 Å after postprocessing. From the initial 3D classification a single class of Nog2$^{post}$ that contained 102,436 particles was also used for further processing. A mask was made surrounding the Rix-complex to subtract core density, this was followed by local 3D classification without re-alignment to select Rix-complex-containing particles and resulted in a single class with 50,091 particles and a final resolution of 2.90 Å after 3D refinement and postprocessing. Only Nog2 was modeled in this map and the GTP was modified to GDP, the reference model that should be used for this map is PDB 6YLH[13].

### Nog2•Tif6•AlF$_4^-$ (*spb1$^{DS2A}$*) dataset

Initial 2D classes were obtained the same way as for the Nog2•Tif6 (*SPB1*) dataset. 790,530 particles were extracted after automated particle picking, binned 4 times and used for 2D classification, selecting 716,018 particles for subsequent 3D classification. Three classes corresponding to Nog2$^{pre}$ were combined and re-extracted at the original pixel size of 1.08 Å, resulting in the total of 417,994 particles. 3D refinement was performed with an imposed symmetry of C1 and resulted in an initial reconstruction with a resolution of 2.78 Å. CTF refinement followed by another cycle of 3D refinement and postprocessing improved the overall resolution to 2.38 Å.

### Nog2•Tif6 (*spb1$^{DS2A, E769K}$*) dataset

Initial data processing was performed as for the Nog2•Tif6 (*SPB1*) dataset. 1,207,583 particles were selected from 2D classification, binned 4 times, and extracted for subsequent 3D classification. Five Nog2$^{pre}$ classes attained the theoretical resolution limit and looked identical (923,524 particles). However, to expedite data processing only one class containing 330,196 particles was selected for further processing. CTF refinement and 3D refinement followed by selection of particles with strong H68-containing density resulted in a total of 320,145 particles. After subsequent polishing, 3D refinement and postprocessing a final map with a resolution of 2.36 Å was obtained.

### Cryo-EM model building and refinement

PDB 3JCT[14] was used as a starting model for the Nog2$^{pre}$ (*SPB1*) map and the refined rebuilt model was used for building and refining into the other Nog2$^{pre}$ maps generated in this study. PDB 2GJA[22] was used as to validate the placement of both the waters and GDP • AlF$_4^-$ in the Nog2 active site in the Nog2$^{pre}$-AlF$_4^-$ (*spb1$^{DS2A}$*) dataset. Nog2 from PDB 6YLH[13] was used as a starting model for Nog2$^{post}$ (*spb1$^{DS2A}$*). All models were built in Coot[41] and refined in PHENIX using phenix.real_space_refine[42].

Figures were generated with UCSF Chimera[43], Chimera X[44], and the PyMOL Molecular Graphics System (Version 2.0 Schödinger, LLC.).

### Time course microscopy methods

Strains YJE001, YJE944 and YJE869 were transformed with pRS316-uL18-GFP or pRS316-uS5-GFP[26,28]. Strains were grown in SCD-URA at 30 °C or 37 °C to saturation and then diluted 1:10 into SCD-URA and grown at 30 °C. At each time point, cells were mounted in 1% low melt agarose and imaged immediately. Fluorescence micrographs were recorded on an inverted Nikon EclipseTi microscope fitted with a 100X Plan Apo objective and an Andor Zyla 5.5 sCMOS camera.

### Purification of Tif6 containing pre-ribosomes

Starters of YJE608 (*SPB1*) and YJE735 (*spb1$^{DS2A}$*) harboring a genomic C-terminal 3xFLAG−3C−2xProteinA tag on Tif6 were grown overnight in YPD to saturation. The following morning 12 L of YPD were inoculated from the starters to an OD$_{600}$ of 0.2 and grown at 30 °C to an OD$_{600}$ of 1.2. Cells were harvested and lysed identically to samples used for cryo-EM, described above. Lysate was cleared in the same way and loaded onto an IgG Sepharose resin (Cytiva). Samples were washed with 10 column volumes of Ribo-buffer A followed by 10 column volumes of Ribo-buffer B. Protein-A tags were cleaved on-column using 3 C protease and eluates were added to 2.0 ml Eppendorf tubes with FLAG resin (Sigma) equilibrated in Ribo-buffer B and incubated on a rocker at 4 °C for 2 h. The slurry was transferred into MicroSpin G-25 Columns (GE Healthcare), washed with 2 ml of Ribo-buffer B, then mixed with 100 µL of Ribo-buffer B enriched with 0.5 mg/ml FLAG peptide and incubated at 4 °C for 30 min. Samples were eluted by gravity flow and concentrated to an A$_{260}$ml$^{-1}$ of 3 (100 nM) using an Amicon Ultra 0.5 ml spin column with a 100 kDa cutoff (Merck Millipore).

### Semi-quantitative mass-spectrometry analysis

3.3 µM of Tif6 containing pre-ribosomes were run on a 4−20% denaturing Gel (BioRad) at 160 V for 7 min and stained with Coomassie Brilliant Blue R-250 (Thermo Fisher). The portion of the gel with protein bands was excised and used for subsequent analysis. Semi-quantitative mass-spectrometry data was obtained by the UTSW Proteomics core, analyzed using Proteome Discoverer 2.4 and searched against the yeast protein database from UniProt. Only high-confidence spectra (FDR ≤ 1%) were considered and are listed in Source Data. Selection of RBFs for our plot was based on their presence in known intermediate structures or well-established presences in specific pre-60S intermediates.

### Reporting summary

Further information on research design is available in the Nature Portfolio Reporting Summary linked to this article.

## Data availability

The cryo-EM density maps and models have been deposited in EMDB and PDB databases under accession codes: EMD-26651 /pdb-7UOO (Nog2$^{pre}$ from the *SPB1* strain), EMD-26689 (Nog2$^{pre}$ 5S rRNP local map), EMD-26703/pdb-7UQZ (Nog2$^{pre}$ from *spb1$^{DS2A}$* strain), EMD-26799 /pdb-7UUI (Nog2$^{post}$ from *spb1$^{DS2A}$* strain), EMD-26686 /pdb-7UQB (Nog2$^{pre}$ with AlF$_4^-$ from *spb1$^{DS2A}$* strain), EMD-26941 /pdb-7V08 (Nog2$^{pre}$ from *spb1$^{DS2A/E769K}$* strain). Strain and plasmid information, cryo-EM micrographs, 2D classes, FSC curves, ResMap plots, and data processing 3D-classification and particle sorting schemes generated in this study are provided in the Supplementary Information/Source Data file. Correspondence and requests for materials should be addressed to J.P.E. Source data are provided with this paper.

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

## Acknowledgements

We thank Daniel Stoddard and Jose Martinez at the UTSW Cryo-Electron Microscopy Facility, funded in part by the CPRIT Core Facility Support Award RP170644, James Chen at the UTSW Structural Biology Lab and Matthew Parker for help with light microscopy. A portion of this research was supported by NIH grant U24GM129547 and performed at the PNCC at OHSU and accessed through EMSL (grid.436923.9), a DOE Office of Science User Facility sponsored by the Office of Biological and Environmental Research, with the assistance of Theo Humphreys. We thank V. Panse for sharing plasmids pRS316-uL18-GFP and pRS316-uS5-GFP. The

authors would also like to acknowledge the Quantitative Light Microscopy Core, a shared resource of the Harold C. Simmons Cancer Center, supported in part by an NCI Cancer Center Support Grant, 1P30 CA142543-01 and the UTSW Proteomics Core. J.P.E. was supported by the Cancer Prevention and Research Institute of Texas (RR150074), the Welch Foundation (I-1897), the UTSW Endowed Scholars Fund and the National Institutes of Health (GM135617-01).

## Author contributions

K.S., V.E.C., C.S.W and J.P.E. conceived the project. K.S., V.E.C., and C.S.W. constructed strains. K.S. and C.S.W. conducted all growth and microscopy assays. V.E.C. and K.S. carried out sample preparations and collected cryo-EM data. V.E.C., K.S. and J.P.E. processed the cryo-EM data. J.P.E., K.S., V.E.C. and C.S.W. wrote the paper. J.P.E. supervised the work.

## Competing interests

The authors declare no competing interests.
