## [Peer Review File · Nature Communications]

rRNA methylation by Spb1 regulates the GTPase activity of Nog2 during 60S ribosomal subunit assemblyREVIEWER COMMENTS

Reviewer #1 (Remarks to the Author):

Sekuklski et al. report cryo-EM structures of a 60S pre-ribosome/Nog2 particle isolated from a catalytic dead sbp1D52A mutant that lacks methylation of the specific base G2922 within 25S rRNA. The structure provides high local resolution of helix 92 (H92) with G2922 and the assembly factor Nog2 in the absence of the G2922 modification. Using ADP-AlF₄ during structural studies they find that unmodified G2922 coordinates water molecules for nucleophilic attack prior GTP hydrolysis. The authors conclude that G2922 methylation by Sbp1 inhibits Nog2 GTPase activity. Further, based on decreased yields of purified particles from sbp1-D52A mutant strain authors reason pre-mature Nog2 GTPase activation, i. e. lack of G2922 methylation disrupts Nog2 interaction with H92, which in turn leads to reduced formation of nucleoplasmic pre-60S particles. The authors attempt to support this model by a genetic rescue experiment of sbp1D52A mutant by Nog2 over-expression and two intragenic suppressor alleles of Sbp1 that presumably improve Nog2 binding to H92.

The structural data presented are of excellent quality. However, this is not a new state of a 60S pre-ribosome, nor does it localize new assembly factors. Therefore, the impact of the determined structure(s) on the field is limited. The main conclusion of the manuscript regarding the function of unmethylated G2922 base remains as a hypothesis that has not been experimentally tested with a clear cut experiment, a standard of the ribosome assembly field. I list specific issues that the authors need to be experimentally address.

Major points

Based of cryo-EM studies the authors claim premature activation of Nog2 GTP hydrolysis in the absence of G2922 base methylation. Since the cryo-EM structure represents specific a snapshot for a subset of particles, the authors should directly test in vitro Nog2-GTPase activation by the unmethylated G2922 within H92 and more importantly Nog2 inactivation by the methylated G2922.

The premature activation by the unmethylated G2922 base raises a pertinent question: how is Nog2 activated under wild-type conditions when G2922 is methylated? This should be discussed.

The authors state that the timing of the GTP hydrolysis is not critical for the final Nog2 release. This contrasts with the previous report (Matsuo et al 2014 Nature) and what is also described in the manuscript. The authors should provide a model/explanation when and how Nog2 is released in WT and sbp1-D52A.

The authors suspect 60S biogenesis defect at early stages prior the 5S RNP rotation. It is problematic to reach this conclusion based only on the reduced yields of Nog2 pre60S particles from *sbp1-D52A* mutant strain. Particle compositional analysis isolated via another nucleolar/nuclear assembly factor (such as Nop7 or Rix1) should be performed to support this claim. For e.g. A semi-quantitative mass spectrometry analyses of such a purified particle (in wild-type and mutant backgrounds) is required to demonstrate an increased levels of early assembly factors and a decreased levels of late nucleoplasmic assembly factors.

The authors attempt to get supportive genetic data through rescue of the growth defect of the *sbp1-D52A* mutant by Nog2 over-expression. The Nog2 GTPase mutant (*nog2G369A*) does not rescue *sbp1-D52A* growth defect. Interestingly, the *nog2G369A* does not show dominant-negative phenotype in this study as previously shown in Matsuo et al., 2014 Nature. Could authors comment on this?

The authors speculate that the underlying mechanism of the *Sbp1*-suppressors increases affinity of Nog2 to the A-loop but not *Sbp1* restored methyltransferase activity – the latter is shown structurally. However, this claim needs to be directly tested to support the suppressor claim/hypothesis.

Overall, the presented data provide little evidence to claim that G2922 methylation provides a kinetic checkpoint for 60S maturation as stated in the abstract. Therefore, the authors broad conclusion in the abstract that “pre-mature activation of Nog2 serves as a template to study RNA trans factors”, is misleading based on the data presented.

Minor point

The authors claim in the last sentence of the paragraph “G2922 directly activates Nog2”: “We believe this represents the first evidence of direct, methylation-dependent activation of a GTPase by an RNA nucleotide”.

As the methylation is inhibitory this sentence needs to be rephrased.

Reviewer #2 (Remarks to the Author):

In this manuscript the authors use cryo-EM and yeast genetics to study the activities of the methyltransferase *Sbp1* and the GTPase *Nog2* during assembly of the large ribosomal subunit in yeast. *Sbp1* specifically methylates G2922 from the A-loop and is an important structural component of pre-ribosomes, however the role of G2922 methylation is unknown. *Nog2* is an essential assembly factor involved in numerous pre-ribosome assembly steps, including anchoring of the 5S rRNP and recruitment

of the export factor Nmd3. The authors hypothesized that given the proximity of mG2922 to the GTPase active site of Nog2, this modification may mediate Nog2 function. To test this hypothesis, they determined a series of high-resolution cryo-EM structures of pre-60S intermediates, which allowed for the visualization of rRNA modifications. The authors observed that the methyl group of mG2922 is near the Nog2 GTPase center. Through use of an Sbp1 mutant they showed that loss of G2922 methylation leads to premature activation of Nog2. The authors propose that methylation is an important checkpoint during ribosome assembly that regulates the GTPase activity of Nog2. Overall, this manuscript provides significant new information on the role of G2922 methylation, however I do have some concerns that the authors need to address:

- The authors determined 4 pre-60S structures with resolutions in the range of 2-3Å. The brief manuscript is understandably focused on Nog2 and Sbp1, however I think it is important that the authors go back and do a more in-depth analysis of these pre-60S structures. While I am sure that the structures are similar to previously determined structures such as PDBID 3JCT, I am curious if any additional information was learned from the increase in resolution. For example, the authors mention the Rix1 complex, but it is not shown in any of the figures. The authors also show density for other visible rRNA modifications, but this is not discussed in the text.
- Nog2 is a “hub protein” interacting with numerous other assembly factors and several regions of the rRNA. Does loss of G2922 methylation disrupt any known Nog2 pre-ribosome interfaces outside of the A-loop?
- Addition of a cartoon schematic or multiple sequence alignment of Sbp1 and Nog2 would be helpful to understand where the mutations are.
- Please label the nucleotides with residue numbers in Extended Data Figure 5.

Reviewer #3 (Remarks to the Author):

The authors use a clever combination of genetics and inhibitors to dissect the conformational intermediates in the activation of Nog2 GTPase. Further, they propose that Nog2 bound to pre60S constitutes a checkpoint in the regulation of the formation of the large subunit. Improvements in data quality over previous structural work permitted the refinement of observations and the determination of mechanistic processes hitherto undefined. This advances the field.

The article is concise, perhaps to the point of making it difficult to read. Figures are pleasing to the eye, but some details, especially in the overall maps are difficult to discern, perhaps due to a choice of colors that do not seem to consider color blind audience.

The quality of the maps seems sufficient to support the author’s conclusions. However, details on the local density and resolution of the crucial portions of the map are unavailable in the manuscript. Moreover, assertions like the following one should be toned down: ...”The high resolution of our maps

allows us to precisely map the coordination network around the Mg²⁺ and K⁺ metal ions in the Nog2 active site in both the GDP and GTP-bound states”... The quality of the maps allowed them to resolve the metal ions and some of the features of the residues coordinating them. This facilitated the inference of the coordination network (which was not mapped) based on prior chemical knowledge.

The choice of scaling of resolution maps (ResMap) does not aid in interpreting the local resolution of the interesting bits of the structures shown. Regrettably, the authors chose not to include atomic models fitted into density maps in the review process. Thus, it becomes almost impossible to judge the quality of the data in the relevant portions, nor many of the assertions presented as atomic representations. It might be a personal preference, but I would have liked to see the planar configuration of tetrafluoroaluminate as a density map.

Some minor points include the use of Absorbance Units to express the concentration of the species prepared for cryo-EM, it would be useful to convert it to molar concentration. Some blotting parameters such as time and fore, although perhaps not critical, should probably be included. Similarly, the number of collected movies is specified for all datasets but not the final number of micrographs used for particle selection.

RE: Point-by-point response for manuscript NCOMMS-22-24117-T

Reviewer #1

Sekuklski et al. report cryo-EM structures of a 60S pre-ribosome/Nog2 particle isolated from a catalytic dead sbp1D52A mutant that lacks methylation of the specific base G2922 within 25S rRNA. The structure provides high local resolution of helix 92 (H92) with G2922 and the assembly factor Nog2 in the absence of the G2922 modification. Using ADP-AIF4 during structural studies they find that unmodified G2922 coordinates water molecules for nucleophilic attack prior GTP hydrolysis. The authors conclude that G2922 methylation by Sbp1 inhibits Nog2 GTPase activity. Further, based on decreased yields of purified particles from sbp1-D52A mutant strain authors reason pre-mature Nog2 GTPase activation, i. e. lack of G2922 methylation disrupts Nog2 interaction with H92, which in turn leads to reduced formation of nucleoplasmic pre-60S particles. The authors attempt to support this model by a genetic rescue experiment of sbp1D52A mutant by Nog2 over-expression and two intragenic suppressor alleles of Sbp1 that presumably improve Nog2 binding to H92.

The structural data presented are of excellent quality. However, this is not a new state of a 60S pre-ribosome, nor does it localize new assembly factors. Therefore, the impact of the determined structure(s) on the field is limited.

While the reviewer is correct in pointing out that the Nog2-pre-rotation state has been previously reported, the increased resolution of our reconstruction has allowed us to improve and extend molecular models for several RPs and RBFs, revealing post-transcriptional modifications and molecular snapshots of enzymatic active sites that we believe will be of general interest to the field as it begins to investigate the mechanistic details of ribosome assembly. We have emphasized this at the beginning of our results section (see also reviewer 2):

Page 2 - Line 11

“While there are no major differences in...number of register shifts and mis-paired rRNA bases as well as improving the completeness of individual RBF and RP models.”

The main conclusion of the manuscript regarding the function of unmethylated G2922 base remains as a hypothesis that has not been experimentally tested with a clear cut experiment, a standard of the ribosome assembly field. I list specific issues that the authors need to be experimentally address.

Major points

Based of cryo-EM studies the authors claim premature activation of Nog2 GTP hydrolysis in the absence of G2922 base methylation. Since the cryo-EM structure represents specific a snapshot for a subset of particles, the authors should directly test *in vitro* Nog2-GTPase activation by the unmethylated G2922 within H92 and more importantly Nog2 inactivation by the methylated G2922.

While it is true that our final cryo-EM reconstructions represent a subset of the initial particle dataset, in the three datasets obtained from the mutant background (*spb1^{D52A}*, *spb1^{D52A}+AlF₄⁻* or *spb1^{D52A/E769K}*), all non-junk particles have GDP in the active site and there isn't a subset of particles with GTP or methylated G2922, the latter being consistent with published results that show no evidence for mG2922 in the *spb1^{D52A}* strain.

We considered testing the effect of H92 on Nog2 GTPase activity *in vitro*. However, Nog2 only has very weak GTPase activity *in vitro* (as measured by Matsuo et. al). All x-ray structures of isolated bacterial K-loop GTPases homologous to Nog2 show that the K-loop is disordered, even in the presence of nucleotide. We observe that Nog2 K-loop organization upon pre-60S binding depends on interactions with H64. Therefore, an isolated H92 is unlikely to induce the proper active site conformation of the enzyme, even if H92 could adopt the proper A-loop structure *in vitro*. We have added a section explaining this concept and have added a panel to Extended Data Fig. 8, comparing isolated and ribosome-bound K-loop orientations:

Page 3, line 21

"These conformational changes... and combine to organize its active site."

Because of these limitations, we chose the alternative approach of reconstituting the Nog2 transition state through the addition of AlF₄⁻ to test the hypothesis that H92 directly modulates the GTPase activity of Nog2 in its proper molecular environment. The similarity of the resulting active site geometry to that of MnmE, solved at high-resolution by the Wittinghofer group, further supports our interpretation.

The premature activation by the unmethylated G2922 base raises a pertinent question: how is Nog2 activated under wild-type conditions when G2922 is methylated? This should be discussed.

We are actively investigating the Nog2 activation mechanism in wild-type strains. We have added a section in our discussing addressing this question:

Page 4, line 39

"Nog2 GTPase activation in wild-type cells

Activation of Nog2 GTPasetrans-activating element."

The authors state that the timing of the GTP hydrolysis is not critical for the final Nog2 release. This contrasts with the previous report (Matsuo et al 2014 Nature) and what is also described in the manuscript. The authors should provide a model/explanation when and how Nog2 is released in WT and *spb1-D52A*.

Nog2 GTP binding and hydrolysis is clearly important for the timing of Nog2 release and our findings are consistent with previous studies of Nog2 catalytic mutants. However, our unique suppressor mutations, which

grow almost like wild-type, suggest that under these circumstances, cells tolerate early GTP hydrolysis by Nog2, uncoupling it from the final release of Nog2. This is a departure from classical small GTPases, where hydrolysis and release are tightly coupled. In our model, hydrolysis is still essential for Nog2 release, but hydrolysis and release can be uncoupled. We have clarified our hypothesis in our expanded discussion:

Page 4, line 39

“Nog2 GTPase activation in wild-type cells

Activation of Nog2 GTPasetrans-activating element.”

The authors suspect 60S biogenesis defect at early stages prior the 5S RNP rotation. It is problematic to reach this conclusion based only on the reduced yields of Nog2 pre60S particles from *spb1-D52A* mutant strain. Particle compositional analysis isolated via another nucleolar/nuclear assembly factor (such as Nop7 or Rix1) should be performed to support this claim. For e.g. A semi-quantitative mass spectrometry analyses of such a purified particle (in wild-type and mutant backgrounds) is required to demonstrate an increased levels of early assembly factors and a decreased levels of late nucleoplasmic assembly factors.

We agree with the reviewer that our model predicts an increase in pre-60S nucleolar intermediates in the *spb1^{D52A}* strain. We also agree that our use of prep yields is flawed, as it will be affected by the relative expression levels of the Nog2. We have therefore removed the section about yields as well as Supplemental Table 3. Instead, as the reviewer suggested, we performed semi-quantitative mass spectrometry analysis of 60S intermediates purified with a dual affinity tag on the RBF Tif6, which is present on pre-60S intermediates from early nucleolar to late cytoplasmic maturation and therefore allows the isolation of a broad spectrum of pre-60S intermediates. This analysis shows the predicted enrichment of nucleolar RBFs and a reduction of late nucleoplasmic and cytoplasmic RBFs, as shown in the new Fig. 5d. The experiment is described in the section:

Page 4, line 18

*“Nucleolar/nuclear accumulation of pre-60S in *spb1^{D52A}* cells”*

Additional evidence of the accumulation of early 60S intermediates comes from the light microscopy experiments described below.

The authors attempt to get supportive genetic data through rescue of the growth defect of the *spb1-D52A* mutant by Nog2 over-expression. The Nog2 GTPase mutant (*nog2G369A*) does not rescue *spb1-D52A* growth defect. Interestingly, the *nog2G369A* does not show dominant-negative phenotype in this study as previously shown in Matsuo et al., 2014 Nature. Could authors comment on this?

Our overexpression system (Gal vs Tet-off) and strain background (BY4741 vs W303) differs from that used by Matsuo et al, which may explain the observed differences. To be absolutely certain, we re-sequenced our overexpression plasmids and repeated the experiment, which confirmed both the *NOG2* dosage suppression and our observation that *nog2^{G369A}* is not dominant-negative in our strain background and overexpression system. We added a sentence to the text to address the observed differences:

Page 3, line 38

“Unlike previous studies,...background and overexpression systems”

The authors speculate that the underlying mechanism of the *Sbp1*-suppressors increases affinity of Nog2 to

the A-loop but not Sbp1 restored methyltransferase activity - the latter is shown structurally. However, this claim needs to be directly tested to support the suppressor claim/hypothesis.

In the absence of the γ -phosphate in the *spb1^{D52A}* strain (with or without suppressors), we see a near total loss of density for H92 bound to Nog2, so our argument is not that the Nog2/H92 interaction is restored. Our proposed model relies on a steric argument, i.e. that the V-shaped helix of Spb1 that contains the mutations physically prevents Nog2 binding. We believe that obtaining quantitative data for Nog2 binding affinity to various intermediates *in vitro* would be challenging and beyond the scope of this paper. We have instead bolstered our argument by showing that the suppressor mutant is able to rescue the *in vivo* accumulation of pre-60S particles in the *spb1^{D52A}* cells (see below and new Fig. 5a, b).

Since our original submission, we have also made an *SPB1* shuffle strain that allowed us to test the suppressor mutations in a more controlled manner. These new data replace our original panel in Fig.4a, showing that the suppressor mutations on their own do not affect growth.

Overall, the presented data provide little evidence to claim that G2922 methylation provides a kinetic checkpoint for 60S maturation as stated in the abstract. Therefore, the authors broad conclusion in the abstract that “pre-mature activation of Nog2 serves as a template to study RNA trans factors”, is misleading based on the data presented.

We agree with the reviewer that our initial paper lacked direct evidence for a kinetic barrier. While both the known temperature dependence of the growth defect in the *spb1^{D52A}* strain and our suppressors are consistent with the kinetic proofreading model, we decided to perform light microscopy experiments with GFP-tagged large and small subunit proteins to directly test our model. A time course of ribosome production shows three findings:

1. Production of large ribosomal subunits is slower in the *spb1^{D52A}* background compared to wild type cells.
2. The biogenesis of small ribosomal subunits is not affected.
3. The suppressor mutant relieves the kinetic block and restores growth even when GTP is prematurely hydrolyzed.

Together, these observations are consistent with the kinetic checkpoint model (which is itself consistent with the proposed function of Nog2) and we believe that our combined data is now sufficient for us to “propose” it in our abstract. We have added a section to describe this experiment and the semi-quantitative mass spectrometry described above:

Page 4, line 18

“Nucleolar/nuclear accumulation of pre-60 in *spb1^{D52A}* cells

Both the suppressor...nucleoplasmic and cytoplasmic maturation (Fig. 5d).”

The reviewer is referencing the following sentence in our abstract: *“Because multiple K-loop GTPases are involved in the assembly of ribosomes and other RNPs, our findings provide a template to study the role of RNA trans factors in modulating the regulatory functions of this important family of enzymes.”* We did not mean to imply that premature activation may be a common feature of all GTPases. Rather, we meant to suggest that high-resolution structural studies, combined with the trapping of transition state intermediates, could elucidate the activation mechanisms of other K-loop GTPases. We have clarified the sentence to read:

Page 1, line 22

“Our approach and findings provide a template to study the GTPase cycles and regulatory factor interactions of the other K-loop GTPases involved in ribosome assembly.”

Minor point

The authors claim in the last sentence of the paragraph “G2922 directly activates Nog2”: “We believe this represents the first evidence of direct, methylation-dependent activation of a GTPase by an RNA nucleotide”. As the methylation is inhibitory this sentence needs to be rephrased.

We agree and have changed the sentence to read:

Page 3, line 5

“We believe this represents the first evidence of direct activation of a GTPase by an RNA nucleotide.”

Reviewer #2

In this manuscript the authors use cryo-EM and yeast genetics to study the activities of the methyltransferase Sbp1 and the GTPase Nog2 during assembly of the large ribosomal subunit in yeast. Sbp1 specifically methylates G2922 from the A-loop and is an important structural component of pre-ribosomes, however the role of G2922 methylation is unknown. Nog2 is an essential assembly factor involved in numerous pre-ribosome assembly steps, including anchoring of the 5S rRNP and recruitment of the export factor Nmd3. The authors hypothesized that given the proximity of mG2922 to the GTPase active site of Nog2, this modification may mediate Nog2 function. To test this hypothesis, they determined a series of high-resolution cryo-EM structures of pre-60S intermediates, which allowed for the visualization of rRNA modifications. The authors observed that the methyl group of mG2922 is near the Nog2 GTPase center. Through use of an Sbp1 mutant they showed that loss of G2922 methylation leads to premature activation of Nog2. The authors propose that methylation is an important checkpoint during ribosome assembly that regulates the GTPase activity of Nog2. Overall, this manuscript provides significant new information on the role of G2922 methylation, however I do have some concerns that the authors need to address:

- The authors determined 4 pre-60S structures with resolutions in the range of 2-3Å. The brief manuscript is understandably focused on Nog2 and Sbp1, however I think it is important that the authors go back and do a more in-depth analysis of these pre-60S structures. While I am sure that the structures are similar to previously determined structures such as PDBID 3JCT, I am curious if any additional information was learned from the increase in resolution. For example, the authors mention the Rix1 complex, but it is not shown in any of the figures. The authors also show density for other visible rRNA modifications, but this is not discussed in the text.

Compared to 3JCT, there are no major differences (i.e. previously unresolved RBFs, etc), and the differences between our 4 structures are restricted to the Nog2/H92 interaction. We have modified that section of the paper to address this:

Page 2, line 11

“While there are no major differences in RBF composition compared to previous structures, the ~0.7Å improvement in the overall resolution of our maps compared to previous reconstructions of Nog2^{pre} enabled us to build an improved atomic model that includes all observable rRNA nucleotide modifications, ordered Mg²⁺ ions and their coordinated waters as well as three ordered Bis-Tris-Propane molecules (Extended Data Table 1 and Extended Data Fig. 5). Our model corrects a number of register shifts and mis-paired rRNA bases as well as improving the completeness of individual RBF and RP models.”

- Nog2 is a “hub protein” interacting with numerous other assembly factors and several regions of the rRNA. Does loss of G2922 methylation disrupt any known Nog2 pre-ribosome interfaces outside of the A-loop?

We observe no significant changes in the other Nog2 interactions due to H92 disengagement, consistent with our proposal that once Nog2 is bound, downstream intermediates are not affected by the absence of the γ -phosphate. We have clarified this:

Page 2, line 19

“There is clear density for the methyl group of mG2922 intercalated in near proximity (~7.2Å) to the γ -phosphate of the GTP (Fig. 1e). In Nog2^{pre} intermediates purified from the spb1^{D52A} strain, Nog2 occupies an identical position and maintains all of its RBF, RP and rRNA interactions, but the Nog2 active site is strikingly different (Fig. 1f).”

- Addition of a cartoon schematic or multiple sequence alignment of Sbp1 and Nog2 would be helpful to understand where the mutations are.

We have added a panel to Figure 4 showing an overall schematic of Spb1 and the position of the mutations, as well as a sequence alignment of the region flanking the mutations in Spb1.

- Please label the nucleotides with residue numbers in Extended Data Figure 5.

We have added identifiers to all the ligands and modified nucleotides depicted in Extended Data Figure 5.

Reviewer #3

The authors use a clever combination of genetics and inhibitors to dissect the conformational intermediates in the activation of Nog2 GTPase. Further, they propose that Nog2 bound to pre60S constitutes a checkpoint in the regulation of the formation of the large subunit. Improvements in data quality over previous structural work permitted the refinement of observations and the determination of mechanistic processes hitherto undefined. This advances the field.

The article is concise, perhaps to the point of making it difficult to read. Figures are pleasing to the eye, but some details, especially in the overall maps are difficult to discern, perhaps due to a choice of colors that do not seem to consider color blind audience.

We have added several passages to our original manuscript to improve the readability of the manuscript, in particular the section discussing new *in vivo* results and our discussion of the activation mechanism in WT cells.

Page 4, line 18

“Nucleolar/nuclear accumulation of pre-60 in spb1^{D52A} cells

Both the suppressor mutants.....nucleoplasmic and cytoplasmic maturation (Fig.5c, d).”

Page 4, line 39

“Nog2 GTPase activation in wild-type cells

Activation of Nog2 GTPase activity...different Nog2 trans-activating element.”

We believe these and other, more minor changes have increased the readability of the manuscript.

We have improved the contrast in Fig1a and 1c as well as all the experimental map depictions to improve the clarity of those panels.

The quality of the maps seems sufficient to support the author's conclusions. However, details on the local density and resolution of the crucial portions of the map are unavailable in the manuscript. Moreover, assertions like the following one should be toned down: ..."The high resolution of our maps allows us to precisely map the coordination network around the Mg²⁺ and K⁺ metal ions in the Nog2 active site in both the GDP and GTP-bound states"... The quality of the maps allowed them to resolve the metal ions and some of the features of the residues coordinating them. This facilitated the inference of the coordination network (which was not mapped) based on prior chemical knowledge.

We agree with the reviewer that interpretation of molecular details is crucially dependent on the quality of the maps. That is precisely why we included panels with experimental map densities in all our structural figures (i.e. Fig. 1e and f, Fig. 2b, Fig. 3b, Fig. 4b and Extended data Fig. 5). We speculate that the map features were not visible on the printouts used by the reviewer, which understandably would make it impossible to accurately assess the soundness of our map interpretation. We have updated these panels to increase the contrast of the densities and will make sure that the final figures allow the densities to be easily discerned in the final version of the figures. As requested, we have also toned our statement to:

Page 2, line 25

"The high resolution of our maps allowed us to infer the coordination network in the active sites of the GDP and GTP-bound states (Fig. 1g, h)"

The choice of scaling of resolution maps (ResMap) does not aid in interpreting the local resolution of the interesting bits of the structures shown. Regrettably, the authors chose not to include atomic models fitted into density maps in the review process. Thus, it becomes almost impossible to judge the quality of the data in the relevant portions, nor many of the assertions presented as atomic representations. It might be a personal preference, but I would have liked to see the planar configuration of tetrafluoroaluminate as a density map.

As explained above, we believe this issue to be due to degradation of our density depictions during the uploading and/or printing of the manuscript. Experimental densities (including of AlF₄⁻ in Fig. 2b) are now depicted with higher contrast.

Some minor points include the use of Absorbance Units to express the concentration of the species prepared for cryo-EM, it would be useful to convert it to molar concentration. Some blotting parameters such as time and fore, although perhaps not critical, should probably be included. Similarly, the number of collected movies is specified for all datasets but not the final number of micrographs used for particle selection.

We reference absorbance units because it is difficult to estimate the extinction coefficient of heterogenous pre-60S mixtures, but have added an approximate molar concentration to our methods. We have also added blotting parameters and the number of micrographs used for particle selection in our methods section.

Sincerely,

REVIEWERS' COMMENTS

Reviewer #1 (Remarks to the Author):

The revised version of the manuscript reads well, and the arguments are easier to follow. While the authors seem hesitant to perform the Nog2 activation experiments, they have addressed my concerns. I support publication of the manuscript. But, my recommendation is to alter the tone of their conclusions. For e.g. for the main message of their work: there is no experiment performed to directly show that G2922 methylation actually modulates Nog2 GTP hydrolysis or that G2922 directly activates Nog2.

Most of their conclusions are inferences/hypothesis drawn from the cryo-EM structures and previous work on GTPases, which is fine, but it would be good to state where there is an inference/hypothesis drawn or where there is supporting experimental data.

Reviewer #2 (Remarks to the Author):

The authors have addressed my previous concerns and I support publication of this manuscript in Nature Communications.

Minor Issues:

1. Please include the PDBIDs for State E2 and NE1 in the Fig. 1 and Fig. 3 legends.
2. The text in Fig 4a is hard to read.
3. The text in Fig. 5c and 5d is also very difficult to read without zooming in on the computer. The information in 5d may be easier to interpret in a table format.

Reviewer #3 (Remarks to the Author):

Thank you for addressing my concerns. I still find the manuscript somewhat difficult to read but it might be a matter of style. The authors emphasize the important structural details. That said, in order for the reviewers to make an assessment of the quality of the work, access to the raw maps (including half-maps) would be of benefit. Also, addition of supplemental movies depicting the structural differences and conclusions described in the text would have improved the clarity of the work.

Addition of a summary paragraph:

The following paragraph was added to the introduction (Page 2, lines 4-11)

“Here, we present cryo-EM reconstructions showing that failure to methylate G2922 causes the premature activation of Nog2 GTPase activity. A structure of the Nog2-GDP-AIF₄⁻ transition state is consistent with a direct role for G2922 in the activation of Nog2. Genetic suppressors and in vivo imaging of the ribosomal protein uL23-GFP indicate that premature GTP hydrolysis prevents the efficient binding of Nog2 to early nucleoplasmic 60S intermediates, resulting in a 60S biogenesis defect. Based on these results, we propose a model in which G2922 methylation levels regulate Nog2 recruitment to the pre-60S near the nucleolar/nucleoplasmic phase boundary, forming a kinetic checkpoint to regulate 60S production.”

Reviewer #1 (Remarks to the Author):

The revised version of the manuscript reads well, and the arguments are easier to follow. While the authors seem hesitant to perform the Nog2 activation experiments, they have addressed my concerns. I support publication of the manuscript. But, my recommendation is to alter the tone of their conclusions. For e.g. for the main message of their work: there is no experiment performed to directly show that G2922 methylation actually modulates Nog2 GTP hydrolysis or that G2922 directly activates Nog2. Most of their conclusions are inferences/hypothesis drawn from the cryo-EM structures and previous work on GTPases, which is fine, but it would be good to state where there is an inference/hypothesis drawn or where there is supporting experimental data.

We have addressed the reviewers' concerns with the following changes:

1. In the abstract (Page 1, line 18), we replaced “shows” with “implicates”.
2. In the new summary paragraph, we include the following sentence (Page 2, lines 5-6) to clarify that the mechanism is inferred from the structures: “A structure of the Nog2-GDP-AIF₄⁻ transition state is consistent with a direct role for G2922 in the activation of Nog2.”
3. We changed the title of the subsection discussing the transition state structure (Page 2, line 41) to read: **“G2922 is positioned to directly activate Nog2”**
4. We qualified our interpretation of the structural similarities to the MnmE structure (Page 3, line 11): “We therefore propose that unmethylated G2922 acts as a *trans*-activating factor,…”

Reviewer #2 (Remarks to the Author):

The authors have addressed my previous concerns and I support publication of this manuscript in Nature Communications.

Minor Issues:

1. Please include the PDBIDs for State E2 and NE1 in the Fig. 1 and Fig. 3 legends.

The PDBIDs have been added.

2. The text in Fig 4a is hard to read.
3. The text in Fig. 5c and 5d is also very difficult to read without zooming in on the computer. The information in 5d may be easier to interpret in a table format.

We have increased the font size in those panels and will make sure that the final versions of the figures are clearly legible.

Reviewer #3 (Remarks to the Author):

Thank you for addressing my concerns. I still find the manuscript somewhat difficult to read but it might be a matter of style. The authors emphasize the important structural details. That said, in order for the reviewers to make an assessment of the quality of the work, access to the raw maps (including half-maps) would be of benefit. Also, addition of supplemental movies depicting the structural differences and conclusions described in the text would have improved the clarity of the work.

We included a movie showing the GTP, GDP-AlF₄, GDP transitions in the active site of Nog2.